# Genomic Diversity Profiling and Breed-Specific Evolutionary Signatures of Selection in Arunachali Yak

**DOI:** 10.3390/genes13020254

**Published:** 2022-01-28

**Authors:** Aneet Kour, Saket Kumar Niranjan, Mohan Malayaperumal, Utsav Surati, Martina Pukhrambam, Jayakumar Sivalingam, Amod Kumar, Mihir Sarkar

**Affiliations:** 1ICAR-National Research Centre on Yak, Dirang 790101, Arunachal Pradesh, India; martina.pukhrambam@icar.gov.in (M.P.); mihir.sarkar@icar.gov.in (M.S.); 2ICAR-National Bureau of Animal Genetic Resources, Karnal 132001, Haryana, India; Saket.Niranjan@icar.gov.in (S.K.N.); amod.kumar@icar.gov.in (A.K.); 3ICAR-National Dairy Research Institute, Karnal 132001, Haryana, India; mohanmalayaperumal@gmail.com (M.M.); utsav.surati@yahoo.com (U.S.); 4ICAR-Directorate of Poultry Research, Hyderabad 500030, Telangana, India; jeyvet@gmail.com

**Keywords:** genomic diversity, selection signatures, double digest restriction-site associated DNA, Arunachali yak

## Abstract

Arunachali yak, the only registered yak breed of India, is crucial for the economic sustainability of pastoralist Monpa community. This study intended to determine the genomic diversity and to identify signatures of selection in the breed. Previously available double digest restriction-site associated DNA (ddRAD) sequencing data of Arunachali yak animals was processed and 99,919 SNPs were considered for further analysis. The genomic diversity profiled based on nucleotide diversity, π (π = 0.041 in 200 bp windows), effective population size, Ne (Ne = 83) and Runs of homozygosity (ROH) (predominance of shorter length ROHs) was found to be optimum. Subsequently, 207 regions were identified to be under selective sweeps through de-correlated composite of multiple signals (DCMS) statistic which combined three individual test statistics viz. π, Tajima’s D and |iHS| in non-overlapping 100 kb windows. Mapping of these regions revealed 611 protein-coding genes including *KIT*, *KITLG*, *CDH12*, *FGG*, *FGA*, *FGB*, *PDGFRA*, *PEAR1*, *STXBP3*, olfactory receptor genes (*OR5K3*, *OR5H6* and *OR1E1*) and taste receptor genes (*TAS2R1*, *TAS2R3* and *TAS2R4*). Functional annotation highlighted that biological processes like platelet aggregation and sensory perception were the most overrepresented and the associated regions could be considered as breed-specific signatures of selection in Arunachali yak. These findings point towards evolutionary role of natural selection in environmental adaptation of Arunachali yak population and provide useful insights for pursuing genome-wide association studies in future.

## 1. Introduction

Mammalian body has evolved certain anatomical (enlarged heart and lungs, shorter tongue), physiological (increased oxygen availability in blood, lower metabolic rate), haematological (modifications in haemoglobin structure to improve oxygen affinity, increased average platelet volume and plasma fibrinogen concentrations) and morphological (small body size, distinct coat features like colour, texture and length) adaptations in order to cope up with the stressors of long-term exposure to cold and hypoxic conditions of high altitudes [1,2,3,4,5,6,7,8]. Most of these adaptations are genetically fixed in highlanders, thus conferring them protection against adverse climatic conditions [9]. Genomic studies have shown that the most common genes responsible for adaptation in highlander species include those involved in hypoxia-inducible factor (HIF) pathway like *EPAS1*, *EGLN1*, *EGLN2*, *VEGF*, *EPO*, *NOS* and *EDN* [10,11]. Further, different populations of the same species may possess varying adaptations in their genomes in response to difference in altitude and vegetation [12,13]. 

Yak (*Bos grunniens*) is a unique species which has been exposed to intensive natural selection during the process of environmental adaptation in order to sustain the harsh environmental extremes of high altitude regions [14,15]. Artificial selection by man and breeding within small geographical territories may have further consolidated certain evolutionary signatures within its genome [16]. Elucidation of these regions subjected to positive selection in yak genome can help in maintaining genetic diversity and unravelling causal variants and genes subjected to selection.

There are around 58,000 yaks in India, around half of which are concentrated in the easternmost state, Arunachal Pradesh. [17]. Arunachali yak (Figure 1), the only registered native yak breed of India (till date) has been traditionally reared for food, fibre, transport and manure by the Monpa pastoralist community of Tawang and West Kameng districts of Arunachal Pradesh. These are medium-sized animals with compact body producing an average milk yield of 1.1–1.6 kg/day and the mean adult body weight in males and females is around 365 kg and 230 kg, respectively [18]. The breed resides in cold and humid climate at an altitude of 2700–4300 m above mean sea level and can sustain a temperature of −40 °C to 10 °C [19]. This is in stark contrast to the yaks found in cold and arid regions of Ladakh (India) and Tibetan Autonomous Region (TAR) of China which dwell at an altitude of 4000–5000 m and 4500–5500 m respectively [20]. Hence, the possibility of distinct selection signatures in the yaks of Arunachal Pradesh cannot be ruled out.

A breed-specific, genome-wide scan for identifying signatures of selection can provide some interesting insights into the evolutionary history of the population and can help to highlight new targets for selection and genetic improvement of the breed. Although the genome-wide SNPs (Single Nucleotide Polymorphisms) present in Arunachali yak breed have been primarily evaluated, a holistic study to explore the positively selected regions in the genome of lone registered yak breed of India is still pending. The present investigation was directed at profiling the genomic diversity and identifying the putative regions under selection in Arunachali yak population based on a genome-wide scan.

## 2. Methods

Double digest restriction-site associated DNA (ddRAD) sequencing technique employs a combination of both rare and frequent cutters or restriction enzymes to generate raw reads after precise size selection [21]. The ddRAD sequencing data for twenty animals of Arunachali yak available in NCBI (BioProject PRJNA577203) was used in the study. The complete analysis was performed on a computer with the following configuration: 12 GB RAM, 256 GB SSD and Microsoft i4 processor.

### 2.1. Quality Control of Sequencing Reads

After demultiplexing to obtain reads for each sample, quality control of the paired-end reads was performed and low quality sequences, sequences with contamination or artifacts were removed to avoid erroneous conclusions. Paired-end FASTQ files with raw reads were analysed in PRINSEQ lite v0.20.4 [22] to trim the adaptor sequences at the ends. The sequences with less than 30 phred quality score at any of the windows were discarded using Stacks [23].

### 2.2. Sequence Alignment and Variant Calling

The processed reads were further aligned to *Bos taurus* reference genome assembly ARS-UCD1.2 [24,25] using Bowtie 2 [26] due to lack of chromosome-wise assembly for yak genome (scaffold-based assembly is available at present). The resulting aligned reads in sequence alignment mode (SAM) format were converted to binary alignment mode (BAM) format (using ‘samtools view’ flag) in SAMtools [27]. Finally, the reads from all the samples were merged into a single binary call format (BCF) file using the flag ‘samtoolsmpileup’.

VCFtools [28] was employed to convert single BCF file into a VCF format which is the ideal format to be used for further analysis. Furthermore, it was used to filter out SNPs at different read depths (RD) viz. RD 2, RD 5 and RD 10 and those at RD 10 were retained for further analysis. SNPs at RD 10 were subjected to quality control (QC) and the ones with minor allele frequency (MAF) lesser than 0.05, call rate (CR) lesser than 90% and *p*-value for Hardy-Weinberg Equilibrium (H-WE) lesser than 0.0001 were removed from the data so as to avoid false positive results.

### 2.3. SNP Annotation and Identification of Deleterious Mutations

Structural and functional annotation of the high quality SNPs was carried out using SnpEff [29]. Further, we identified the deleterious mutations in the population based on SIFT (Sorting Intolerant from Tolerant) scores using VEP [30]. The mutations with SIFT score ≤ 0.05 was adjudged as deleterious to protein function and subsequently, were mapped to their location in the protein-coding genes. The obtained gene list was analysed in PANTHER [31] for statistical overrepresentation of gene ontology (GO) terms.

### 2.4. Genomic Diversity Estimation

Three indicators of genomic diversity-nucleotide diversity (π), effective population size (N_e_) and runs of homozygosity (ROH) were considered to assess genetic variability in the population.

Π as a measure of genetic divergence within population was estimated in 100 kb windows of the genome using “window-pi” flag in VCFtools [28] and an output file with the suffix “Windowed.pi” was generated. In order to calculate the average π for the whole population, 200 bp window was considered for more accurate estimate.

SNeP package in R [32] was used to determine N_e_ and trends in N_e_ trajectories on the basis of relationship between r^2^, N_e_ and recombination rate (c). A ped file was provided as an input in the package and svedf [33] was the mapping function used. The output file revealed the historical N_e_ estimates and genome-wide linkage disequilibrium (LD) patterns.

Additionally, ROH and genomic inbreeding coefficient based on ROH (F_ROH_) in the population were investigated using the consecutive method in detectRUNS package in R [34]. ROH pattern was evaluated in 0–2 Mb, 2–4 Mb, 4–8 Mb and >8 Mb classes.

### 2.5. Selective Sweep Identification

Selective sweep in the population were identified based on de-correlated composite of multiple signals (DCMS) which is a composite measure of selection combining the power of different individual statistics. We used three statistical parameters viz., allele-frequency spectrum-based methods like nucleotide diversity (π) and Tajima’s D and haplotype-based methods like integrated haplotype score (iHS) to estimate the combined DCMS scores.

Similar to the π estimates (calculated earlier), the Tajima’s D was also estimated in windows of 100 kb using “TajimaD” flag in the VCFtools package. For calculating iHS scores, haplotype phasing was performed in Beagle [35] to determine the individual haplotypes. Thereafter, iHS scores for 100 kb windows were obtained using selscan package [36]. Un-standardised iHS scores were normalised using ‘norm’ flag. Normalised iHS scores within each 100 kb window were calculated.

Finally, all three parameters for each 100 kb window were fed into the MINOTAUR package in R [37] to calculate DCMS score for each window. First of all, genome-wide rank-based *p*-values were generated for each of three statistics using “stat_to_pvalue” function in the MINOTAUR package. Afterwards, covariance matrix was constructed based on 50,000 randomly sampled SNPs (with α = 0.75) using “CovNAMcd” function in rrcovNA package in R [38]. This matrix was used to adjust for correlation among the statistics and was used in obtaining a DCMS statistic (using ‘DCMS’ function) in MINOTAUR package. The resulting statistic was fitted into a normal distribution using a robust linear model in MASS package in R [39]. Finally, the fitted DCMS scores were used as input along with the mean (µ) and standard deviation (SD) of the fitted model to calculate the *p*-values based on DCMS scores using ’pnorm‘ function in R. The SNP windows with *p* values < 0.01 were considered as ‘significant’ signatures of selection.

### 2.6. Mapping of Selective Sweeps and Gene Ontology

Based on the marker density and r^2^ between adjacent SNPs, LD decay was calculated using GAPIT package in R [40,41]. SNP windows with significant *p*-values were extended by a certain distance (both upstream and downstream) based on LD decay. This was used to scan the genome for protein-coding genes based on *Bos taurus* genome assembly ARS-UCD 1.2 in UCSC Genome Browser [24,25].

### 2.7. Validation Based on QTL Mapping

In order to validate our findings, putative regions under selective sweep were examined for the presence of quantitative trait loci (QTL) based on Cattle QTLdb (https://www.animalgenome.org/cgi-bin/QTLdb/BT/index, accessed on 10 May 2021) [42] and overlaps between the selected regions and known QTL regions were assessed.

### 2.8. Gene Ontology

Gene ontology (GO) and statistical overrepresentation of GO terms was done in PANTHER [31] by providing the gene list from the putative regions under selective sweep as an input and *Bos taurus* annotation file as a background. A false discovery rate (FDR) value of ≤0.02 was set as the threshold for significance of GO terms.

## 3. Results

### 3.1. SNP Annotation

A total of 96.46 million raw reads of Arunachali yak were obtained for the analysis. *SphI* and *MluCI* restriction enzymes were used for creating reduced representation libraries of the samples. Thus, the sequences lacking the restriction cut sites for both the enzymes and those with phred score (base quality score) lesser than 30 were removed. Finally, 89.95 million good quality reads which constituted 93.25% of the total raw reads were generated after processing.

This was followed by alignment of the filtered good quality reads with *Bos taurus* reference genome assembly ARS-UCD 1.2. The average alignment rate across the sample datasets was 97.71% with >85% alignment rate for all the samples.

The variants (SNPs and Indels) were generated at different read depths viz., RD 2, RD 5 and RD 10. (Table 1) SNPs at RD 10 were considered for further analysis and were subjected to additional QC criteria like MAF > 0.05, CR > 90% and *p*-value for H-WE > 0.0001. Only 99,919 SNPs could pass these stringent criteria and were considered for further analysis.

On structural and functional annotation of SNPs, it was revealed that the total number of effects were 157,173 and around 48% of these were present in the intronic region followed by 41% in the intergenic region. Only 0.63% of the total effects could be localised to the exonic region. The distribution of SNP effects in different regions of the genome is presented in Figure 2.

A total of 68.5% (623) of the variant effects were found to be silent causing no change in the amino acid synthesised, whereas 31.5% (286) were found to be mis-sense leading to the synthesis of a different amino acid. The missense: silent ratio was 0.459. In line with our expectations, the number of transitions (Ts) were more than twice the number of transversions (Tv) i.e., Ts/Tv ratio was 2.879.

With respect to the impact of SNP effects, highly deleterious were about 0.001% (2), low and moderately deleterious constituted 0.5% (730) and 0.2% (286) respectively whereas modifiers were a humongous 99.4% (156,155) of the total effects. The two highly deleterious mutations were located in the non-coding region.

### 3.2. Identification of Deleterious Mutations

A total of 166 SNPs with mis-sense deleterious mutations and SIFT score ≤ 0.05 were mapped to their genomic locations and a gene list with 106 genes was generated (Appendix A). These included those involved in reproduction (*FANCD2*, *ADAM18*, *NUF2*), immunity (*TET2*, *GPR33*), development (*TET2*, *TBX19*) and signalling (*STIL*, *VRK2*, *HTR4*, *ADGRA3*) processes. On functional annotation of this gene list, no statistically significant results were found for GO terms related to biological processes or pathways which reflected that the mutations were not deleterious enough to effect any change in protein function.

### 3.3. Genomic Diversity Estimation

Genomic diversity measured using π with 200 bp windows across the genome was found to be 0.041 indicating sufficient variability in Arunachali yak population.

N_e_ for the most recent generation (i.e., 13 generations ago) was estimated to be 83. However, a serious declining trend in N_e_ was emerging in the past more than 100 generations which is depicted in Figure 3 (and Appendix A). In order to investigate the cause for the drastic reduction in N_e_ over the generations, ROH were calculated. On an average, there were a total of 371 runs while the average length of ROH in the whole population was 875 kb (minimum length was 11 kb while the maximum length was 9 Mb) (Appendix A). The average genomic inbreeding coefficient estimated based on ROH (F_ROH_) was 0.134. The estimates of the number of ROH and F_ROH_ in the population are presented in Appendix A, whereas F_ROH_ estimates across different chromosomes in the population are depicted in Appendix A. To draw meaningful conclusions from the length of ROH patterns, the ROH were categorised based on: 0–2 Mb, 2–4 Mb, 4–8 Mb and >8 Mb lengths. A high proportion of ROH (94.33%) were predominantly falling in the short category i.e., 0–2 Mb followed by 4.66% in 2–4 Mb and 0.96% in 4–8 Mb category. Only 0.03% of ROH belonged to the longer length (>8 Mb) category in the population. The F_ROH_ estimates for the different categories of ROH are elucidated in Appendix A.

### 3.4. Selective Sweep Identification

The individual statistics related to allele frequency spectrum-based methods (like Tajima’s D and π) for identification of selective sweep were calculated in non-overlapping sliding window of 100 kb each. Similarly for haplotype-based methods like iHS, the phased haplotype file was used and the unstandardised iHS values were computed individually for each SNP location across each *Bos taurus* autosome (BTA). Normalised iHS (|iHS|) scores for each 100 kb window were estimated by averaging iHS scores for the SNPs present within that window. The plot for normalised iHS scores against SNP positions is shown in Figure 4. The resulting values for three individual test statistics viz., Tajima’s D, π and iHS at each 100 kb window across the genome were combined by decomposition of *p*-values for each of the test statistics.

DCMS statistics were calculated individually for each BTA in non-overlapping 100 kb SNP windows. On conversion of scores to *p*-values, 207 regions were found to be putatively under selection (*p* < 0.01).

Linkage disequilibrium (LD) decay report was generated to reveal the extent of LD in the genome and to identify the candidate selection regions more comprehensively. It was found that r^2^ value decayed below 0.2 after 200 kb distance between the adjacent SNPs (Figure 5).

Hence, the identified region was extended 200 kb upstream and downstream to unveil the genes in the extended region. These putatively selected regions encompassed 611 protein-coding genes in the identified sweep positions. BTA7 had the maximum number of regions subjected to positive selection (22) whereas BTA24, 25, 28 and 29 had minimum number of selective sweeps (2 each). The detailed list of the selective sweep regions along with the number of variants and associated genes is presented in Appendix A.

### 3.5. Functional Annotation of the Selected Regions

The retrieved protein-coding genes included those involved in reproduction (*SPATA22*, *OXTR*, *M1AP*), growth (*SEMA3D*, *SEMA5A*, *SEMA4F*, *SEMA3A*, *SLIT2*), immunity (*SKINT1*, *KIT*, *TCAM1*, *OASL*, *TXK*, *TEC*) and behaviour (*PRLH*) etc. Interestingly, several genes regulating climatic adaptation were also found in the sweep regions including *IGF1R* for body size, *KIT* and *KITLG* genes for pigmentation, *CDH12* for subcutaneous fat, *OR5K3*, *OR5H6* and *OR1E1* genes for olfaction, *TAS2R1*, *TAS2R3* and *TAS2R4* genes for taste and *FGG*, *FGA*, *FGB*, *PDGFRA*, *PEAR1*, *STXBP3* genes for platelet aggregation and activation.

Subsequently, in order to perform further phenotypic annotation and validation of our findings, we identified QTL from the CattleQTL database that overlapped with the selected regions (Appendix A). The putative regions of selection found in our study overlapped with QTLs for growth (body weight, dry matter intake, average daily gain etc.), reproduction (conception rate, heifer pregnancy rate, fertility index, calving ease, sperm motility, twinning etc.), morphological adaptation (eye pigmentation, facial area pigmentation, coat colour, coat texture, degree of spotting, white spotting etc.) and physiological adaptation (lung percentage, lung weight, cold tolerance, haematocrit, subcutaneous fat, red blood cell distribution width, packed cell volume (PCV) variance, final packed RBC volume, methane production, body temperature, respiratory rate etc.,). QTLs mapped in relation to climatic adaptation traits are described in Table 2.

Functional annotation of the all the genes in the list was performed to highlight the statistical overrepresentation of the GO terms (FDR ≤ 0.02). GO terms related to platelet aggregation (GO:0070527) and detection of chemical stimulus involved in sensory perception of smell (GO:0050911) were found to be highly significant (FDR ≤ 0.02). Other related parent GO terms like platelet activation (GO:0030168), detection of chemical stimulus involved in sensory perception (GO:0050907) and sensory perception of smell (GO:0007608) were also significantly overrepresented in the analysis (Table 3). The complete description of the selective sweep region along with the significant biological processes is presented in Figure 6.

Important genes identified in relation to these processes included: olfactory receptor (OR) genes like *OR5K3*, *OR5H6* and *OR1E1* initiating a neuronal response for triggering sensation of smell [43] and taste receptor genes like *TAS2R1* (Taste 2 Receptor Member 1), *TAS2R3* (Taste 2 Receptor Member 3) and *TAS2R4* (Taste 2 Receptor Member 4) which encode for bitter taste perception [44]. Additionally, as a part of physiological adaptations, fibrinogen genes like *FGG* (Gamma), *FGA* (Alpha) and *FGB* (Beta) along with *PDGFRA* (Platelet Derived Growth Factor Receptor Alpha) gene facilitate platelet aggregation and connective tissue remodelling, thus facilitating wound healing [45,46]. Further, *STXBP3* (Syntaxin Binding Protein 3) and *PEAR1* (Platelet Endothelial Aggregation Receptor 1) genes act to induce platelet activation and secretion [47,48]. 

## 4. Discussion

The genome of Arunachali yak, the lone registered yak breed of India (till date) has evolved majorly as a result of adaptations in cold and humid environments of Arunachal Pradesh and also due to selection efforts of *Monpa* pastoralists of the region. Small and scattered herd size threatens to introduce inbreeding in the population [49]. Knowledge of genetic diversity of a population is critical for genetic improvement of economic traits and serves as an important guide to update the breeding goals and plans, as per the need and situation [50]. Complementarily, identification of selection signatures or adapted genotypes may reveal new targets for selection and may lead to better informed breeding decisions. This will lead to overall improvement in animal productivity and fitness and will contribute greatly to the economic and food security of the pastoralists [51,52]. Hence, genomic diversity estimation and identification of selective sweeps are the twin preliminary objectives of any genetic improvement programme.

### 4.1. Genomic Diversity in Arunachali Yak

In Arunachali yak, the genomic diversity profile was generated using three parameters viz., π, Ne and ROH. The average value of π in the population was 0.041 in 200 bp windows across the genome. This indicated sufficient genomic diversity in the population considering the narrow window size. Comparatively lower (0.0011 in 100 kb windows sliding in 10 kb steps) estimates of π have been reported in native yaks of China [53], possibly due to long-term artificial selection and breeding interventions going on in the yak breeds of the country [54]. In a previous study, higher estimates (π = 0.3058) were reported from the same samples of Arunachali yak [55]. This variation may be attributed to alignment of reads to *Bos mutus* reference genome and a different (probably larger) window size (not mentioned in their study) used to estimate genomic divergence. Further, Sharma et al. (2018) [49] found some evidence of inbreeding and lower genetic diversity in Arunachali yak population based on microsatellite markers. The contrasting findings may have arisen due to the lesser dense cattle-specific microsatellite markers used in the study.

The effective population size was found to be optimum (83) for the most recent generation (i.e., 13 generations ago). The general consensus based on long-term selection experiments suggests that N_e_ of 50–100 is viable for long-term survival of the population [56,57]. Bull rotation practices of pastoralists and chance mating with diverse populations like wild yaks during their course of migration may actually be factors influencing the effective population size in Arunachali yaks. Yet the historical trends indicate that a serious decline in N_e_ has been witnessed over the past more than 100 generations. This calls for persistent efforts to increase the effective population size to greater than 100 in the population in order to maintain the breed for eternity [56]. Nonetheless, errors in N_e_ estimation owing to small sample size used in the study cannot be ruled out [58].

Higher prevalence of shorter and medium length ROH (0–2 Mb and 2–4 Mb) than longer ROHs (4–8 Mb and >8 Mb) in the population was indicative of past inbreeding and shared ancestry between the parents long ago or some population bottleneck in the past. Absence of longer ROHs also indicates optimum genetic diversity and that there has been almost no recent inbreeding in the population.

Hence, the genomic diversity profiling revealed that inbreeding is not a big threat in the breed, as of now and there is a potential for genetic improvement of the population by exploiting the genetic variability. This can be further corroborated with a rising population trend of the breed over the years, sound breeding practices of yak pastoralists (like bull exchange and bull rotation) and possible introgression of wild yak during migration. However, planned mating programme in the breed is the need of the hour to ensure its continual viability and sustainability.

### 4.2. Genomic Regions of Adaptive Change in Arunachali Yak

Predominance of shorter ROHs hinted at the distant demographic and selective events which resulted in repeated fragmentation of chromosomal segments due to recombination. So, the identification of putative regions under selective sweeps or signatures of selection was important to bring further clarity. We used composite measure of selection like DCMS to highlight the selective sweeps in the population. As compared to individual test statistics, DCMS can provide an unbiased and more precise criterion to identify genetic variants under selection by improving signal to noise ratio [59]. Consequently, candidate genes related to the putative selective sweeps can be identified with greater power and accuracy.

GO analysis showed that biological processes associated with adaptation like detection of chemical stimulus involved in sensory perception of smell (GO:0050911), platelet aggregation (GO:0070527) and the related parent GO terms were significantly overrepresented (FDR ≤ 0.02). Olfactory sensation is one of the most genetically evolved physiological adaptation in ruminants at high altitude, and olfactory receptor (OR) genes were highly enriched for hypoxia response in yaks showing evidence of positive selection [10,13,60]. In order to adapt to highly difficult environmental conditions and in response to the diversity in the distribution of vegetation at different altitudes where yaks thrive, OR genes have been highly selected and have undergone rapid evolution during domestication [60]. Moreover, these genes have been implicated in animal adaptation by promoting growth and development of hair follicles [61]. 

Exposure to high altitude and hypoxic conditions also induces platelet hyperreactivity, leading to enhanced platelet adhesion, activation and aggregation [62,63]. Increased platelet function and fibrinogen levels have been documented as an important component of body’s response to chronic hypobaric hypoxia conditions in high altitudes. In a study [64] related to divergent climatic adaptation in yaks and cattle, platelet activation was found to be an enriched biological process encompassing *FGG* and *FGA* (genes also identified in our study). Furthermore, *PDGFRA* (platelet-derived growth factor receptor, alpha polypeptide) gene has been identified to be putatively under selective sweep in Kholmogor and Yaroslavl cattle breeds residing in high altitude cold and harsh climatic region of Russia [65].

Other identified (but not statistically overrepresented) genes in the sweep regions included: *KIT* (KIT proto-oncogene, receptor tyrosine kinase) and *KITLG* (KIT ligand) which are known as key genes involved in the pathway for coat pigmentation [66,67]; *PRLH* (prolactin releasing hormone) gene is known to regulate food intake by relaying satiety signals consequently, influencing feeding behaviour [68]; *IGF1R* (insulin like growth factor 1 receptor) gene mediating the action of *IGF1* gene, thus stimulating body growth [69]. Immune-response related genes like *SKINT1* (selection and upkeep of intraepithelial T cells protein 1), *TCAM1* (testicular cell adhesion molecule 1), *OASL* (2′-5′oligoadenylate synthetase like), *TXK* (TXK tyrosine kinase) and *TEC* (tyrosine protein kinase) were identified. *SKINT1* mediates T-cell differentiation in thymus [70] whereas *TCAM1* gene facilitates immune response during meiosis [71]. *OASL* gene induces innate immunity in response to viral attack [72] while *TXK* and *TEC* genes further contribute to adaptive immune response of the body [7,73]. QTL analysis further validated the existence of QTLs for growth, immunity and adaptation traits overlapping with the putative selective sweeps identified in our study. Most of the regions identified to be positively selected in yaks have their significance in environmental adaptation including physiological modifications, coat colour and skin pigmentation, olfactory sensation, immunity and immune response and hypoxia-related adaptations [10,13,74]. Majority of these pathways like the HIF (hypoxia-inducible factor) pathway have been related to environmental information processing, environmental adaptations, organismal systems and metabolism [74,75]. Most of the species residing at higher altitudes seem to have undergone convergent evolution with respect to genes present in HIF (hypoxia-inducible factor) pathway like *EGLN1*, *EGLN2*, *EGLN3* and *EPAS1* [10]. However, we did not observe any evidence of positive selection for hypoxia-associated genes in our study. It may be due to the fact that yak habitats and vegetation differ greatly within yak-rearing states of India. Trans-Himalayan states like Jammu and Kashmir and Himachal Pradesh (and also, Tibetan region) encounter an extremely cold and arid climate, whereas Arunachal Pradesh, Sikkim and Uttarakhand experience cold as well as humid climate throughout the year [76]. It is further substantiated by the statement that hypoxia-related genes formed an important component for adaptation in the trans-Himalayan region of Ladakh [77]. Thus, it can be safely assumed that adaptations resulting from natural selection may also vary between the yaks found in different climatic and vegetative conditions. In fact, studies such as the present one are indispensable to reveal the breed-specific signals or divergence signals in order to shed light on the ‘breed signatures’ [78].

Our findings of selective sweeps in Arunachali yak are mostly a reiteration of the earlier studies for identifying selection signatures in yaks. However, absence of selection signals for important physiological adaptations related to hypoxia in the breed is reflective of divergent selection within the Himalayan landscape due to differing habitats and ecological conditions prevailing in the specific regions. Future studies with larger sample sizes can unveil more interesting insights for further carrying out genome-wide association studies.

## 5. Conclusions

This study concluded that there was optimum genomic diversity in Arunachali yak breed and genes like *FGG*, *FGA*, *FGB*, *PDGFRA*, *PEAR1*, *STXBP3* and *OR5K3*, *OR5H6*, *OR1E1*, *TAS2R1*, *TAS2R3* and *TAS2R4* have been subjected to strong positive selection for adaptation in the breed. Presence of divergent selection signals further paves the way for identifying breed signatures and designing a genome-wide association study (GWAS) for improving animal productivity and fitness in future. 

## Figures and Tables

**Figure 1 genes-13-00254-f001:**
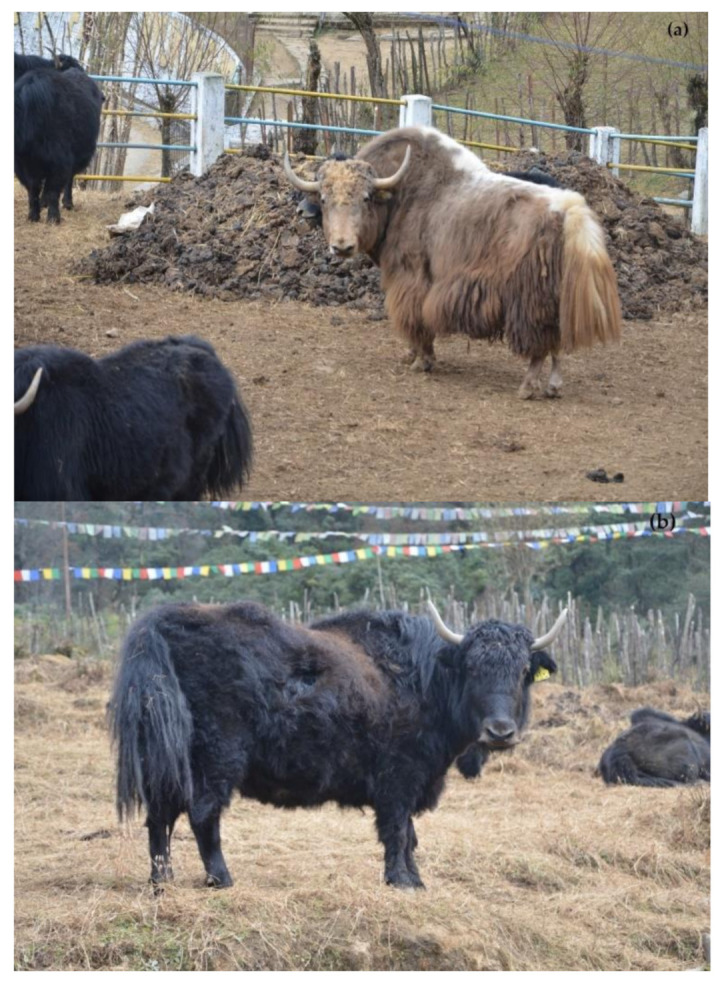
Arunachali yak. (**a**) Arunachali yak male (**b**) Arunachali yak female.

**Figure 2 genes-13-00254-f002:**
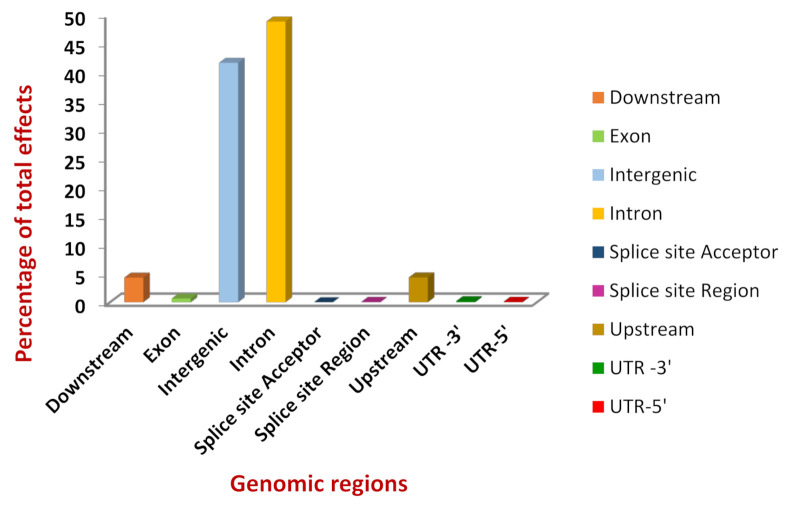
Diagrammatic representation of Single Nucleotide Polymorphism (SNP) effect distribution in the genome of Arunachali yak.

**Figure 3 genes-13-00254-f003:**
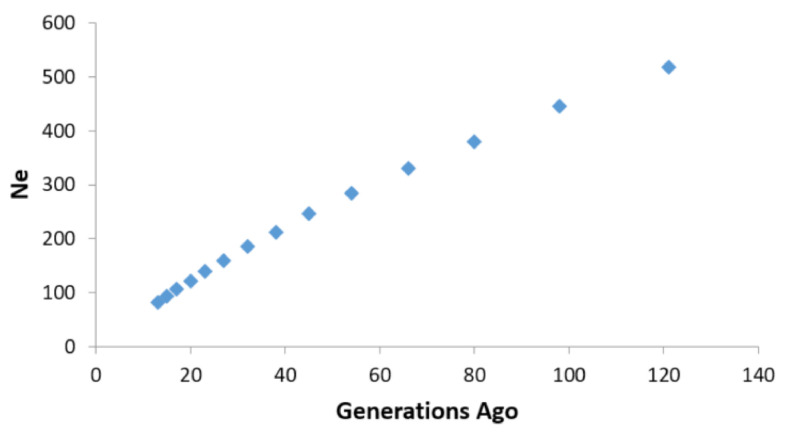
Scatter plot showing N_e_ over the generations (starting from >100 generations ago) based on LD.

**Figure 4 genes-13-00254-f004:**
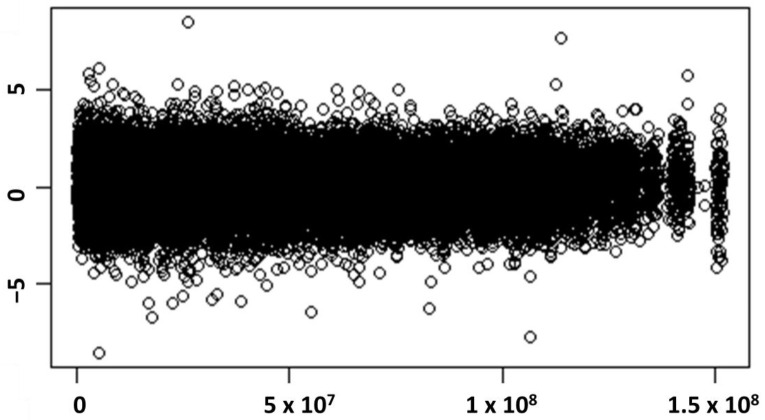
Plot for normalised iHS scores (*Y*-axis) against SNP position (*X*-axis) across the genome.

**Figure 5 genes-13-00254-f005:**
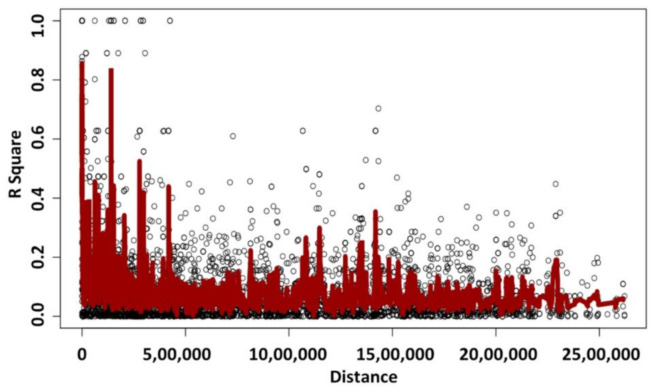
LD decay based on marker density and r^2^ in the population.

**Figure 6 genes-13-00254-f006:**
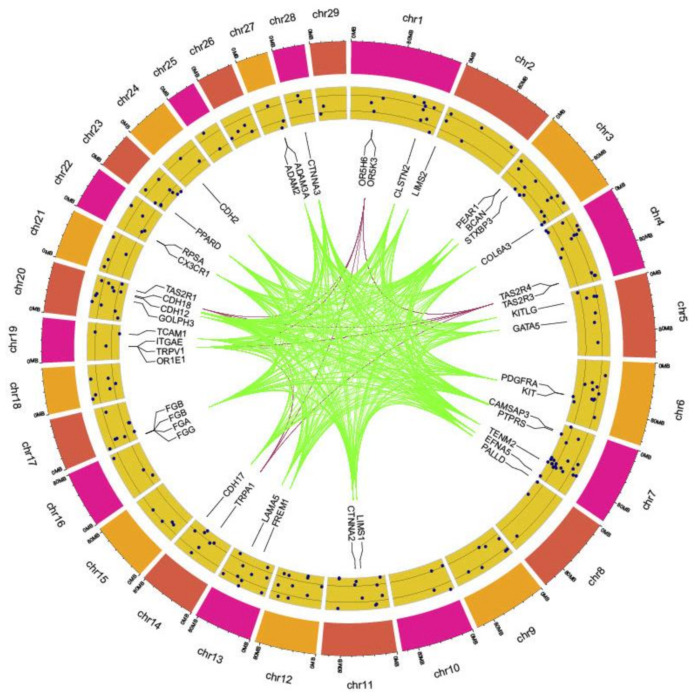
A circos plot depicting putative regions of selective sweep. Outer-Inner: *Bos taurus* autosomes (BTAs); SNP windows; genes associated with a significant biological process with pink colour showing genes involved in olfaction and green colour depicting genes involved in platelet aggregation.

**Table 1 genes-13-00254-t001:** Total variants (including Single Nucleotide Polymorphism (SNP) and Insertion-Deletions (Indels)) at different read depths.

	RD 2	RD 5	RD 10
Total variants	755,786	681,556	634,582
SNPs	701,036	631,816	588,573
Indels	54,750	49,740	46,009

**Table 2 genes-13-00254-t002:** QTLs related to important traits mapped within the selective sweep regions.

Traits	BTA	SNP Window (bp)	QTL Region (bp)	QTL ID
Lung percentage	2	44,800,001–45,300,000	3,033,483–67,663,650	12152
Lung weight	22	12,300,001–12,800,000	10,166,113–34,111,868	12164
Kidney, pelvic and heart fat percentage	2	7,600,001–8,100,000	5,704,346–14,389,811	4857
	3	43,900,001–44,400,000	37,279,925–50,679,768	1352
	6	24,500,001–25,000,000	3,390,665–51,352,653	12153
	7	1,900,0001–19,500,000	7,081,101–22,650,129	4866
	11	54,300,001–54,800,000	20,106,495–66,060,688	15732
	16	72,800,001–73,300,000	72,931,855–72,931,895	152037
Subcutaneous fat	14	35,300,001–35,800,000	31,219,729–43,140,076	20704
	16	31,900,001–32,400,000	31,970,184–31,970,224	157073
	19	47,600,001–48,100,000	47,922,295–48,033,350	18940
Cold tolerance	7	19,000,001–19,500,000	17,086,793–88,971,675	31181
	25	22,800,001–23,300,000	22,013,058–22,964,883	31197
Heat tolerance	12	24,800,001–25,300,000	25,066,554–25,986,217	31189
Haematocrit	11	9,400,001–10,200,000	9,016,028–9,945,427	213425
PCV Variance	17	12,100,001–12,600,000	4,092,903–14,340,160	10533
	20	16,700,001–17,200,000	12,158,768–22,679,451	10538
Final packed RBC volume	17	12,100,001–12,600,000	4,092,903–14,340,160	10534
Percent decrease in PCV up to day 100 after challenge	13	45,900,001–46,400,000	17,709,118–53,561,417	10525
Red blood cell distribution width	15	15,000,001–15,500,000	14,272,339–15,253,334	213481
Haemoglobin	7	1,000,001–1,500,000	971,984–1,887,948	213432
Mean corpuscular haemoglobin concentration	20	41,000,001–41,500,000	41,051,481–42,052,137	213445
Methane production	20	63,300,001–64,300,000	63,407,165–63,407,205	165056
Respiratory rate	24	28,500,001–29,000,000	28,907,134–28,907,174	57040
Eye area pigmentation	6	69,600,001–70,300,000	69,807,007–69,807,047	37348
	4	35,600,001–36,100,000	35,939,851–35,939,891	37346
	5	17,500,001–18,300,000	18,206,797–18,206,837	21151
Facial pigmentation	6	69,600,001–70,300,000	69,807,007–69,807,047	37364
Degree of spotting	18	8,300,001–8,800,000	8,591,333–10,117,552	125378
Coat colour	18	20,100,001–20,600,000	20,060,029–20,212,668	6270
Coat texture	20	37,000,001–37,500,000	37,179,938–37,179,978	32197
White spotting	22	41,700,001–42,200,000	42,111,442–42,111,482	166867

**Table 3 genes-13-00254-t003:** Significant GO terms related to various biological processes (FDR ≤ 0.02).

GO Biological Process	No. of Genes	Fold Enrichment	Raw *p*-Value	FDR
Platelet aggregation	7	11.19	1.02 × 10^5^	2.09 × 10^2^
■ Platelet activation ■ Homotypic cell-cell adhesion ■ Cell-cell adhesion ■ Cell adhesion ■ Biological adhesion	9	7.35	1.08 × 10^5^	1.94 × 10^2^
	8	9.30	7.52 × 10^6^	2.16 × 10^2^
	28	2.99	7.97 × 10^7^	3.82 × 10^3^
	38	2.20	1.34 × 10^5^	1.93 × 10^2^
	38	2.19	1.44 × 10^5^	1.88 × 10^2^
Detection of chemical stimulus involved in sensory perception of smell	3	11	8.38 × 10^9^	6.02 × 10^5^
■ Detection of chemical stimulus involved in sensory perception ■ Detection of chemical stimulus ■ Sensory perception of chemical stimulus ■ Sensory perception of smell	8	28	1.16 × 10^5^	1.85 × 10^2^
	8	28	8.51 × 10^6^	2.04 × 10^2^
	8	27	4.41 × 10^6^	1.58 × 10^2^
	3	11	5.85 × 10^9^	8.41 × 10^5^

## Data Availability

The datasets related to this publication were generated by ICAR-NBAGR, Karnal and can be accessed through NCBI (BioProject PRJNA577203).

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
