# Peer review of "Genomic Diversity Profiling and Breed-Specific Evolutionary Signatures of Selection in Arunachali Yak"

_genes, 2022, doi:10.3390/genes13020254_

Round 1

Reviewer 1 Report

The manuscript is very well written in style. The methods and results are nicely explained. Authors are greatly appreciated for their contribution. Although data set is small and species is rare, the bioinformatics analysis and use of R and other programs is very good.  Few minor edits are as follows:-
Lines 17-18. Provide results of π, Ne and ROH here. 
Lines 40-41. Only 2-3 references may be good enough to make a point. 
Line 70. Pleas add performance of this breed in terms of milk yield, adult weight, longevity etc very briefly in 2-3 sentences and also provide photos (adult male and female)
Line 80. Explain very briefly what is ddRAD sequencing data and how it is obtained?
Lines -83-84. Please add "is" before available.

Reviewer 2 Report

Knowledge of genetic diversity of a population is critical for genetic improvement of economic traits. The present investigation was directed at profiling the genomic diversity and identifying the putative regions subjected to selection in Arunachali yak population based on a genome-wide scan. The research significance of the paper is obvious, but there is still a lot of room for improvement in the presentation of the results and discussion. It is recommended to simplify and accurately describe the results, beautify the paper charts and pay attention to details.

Round 2

Reviewer 2 Report

The authors addressed the suggestions

Author Response

Needful done